# Production and Characterization of Graphene Oxide Surfaces against Uropathogens

Samuel Belo [1,2,†], Francisca Sousa-Cardoso [1,2,†], Rita Teixeira-Santos [1,2], Luciana C. Gomes [1,2], Rita Vieira [1,2], Jelmer Sjollema [3], Olívia S. G. P. Soares [2,4] and Filipe J. Mergulhão [1,2,*]

[1] LEPABE—Laboratory for Process Engineering, Environment, Biotechnology and Energy, Faculty of Engineering, University of Porto, Rua Dr. Roberto Frias, 4200-465 Porto, Portugal; up201608956@edu.fe.up.pt (S.B.); mfcardoso@fe.up.pt (F.S.-C.); ritadtsantos@fe.up.pt (R.T.-S.); luciana.gomes@fe.up.pt (L.C.G.); up201603193@edu.fe.up.pt (R.V.)

[2] ALiCE—Associate Laboratory in Chemical Engineering, Faculty of Engineering, University of Porto, Rua Dr. Roberto Frias, 4200-465 Porto, Portugal; osgps@fe.up.pt

[3] Department of Biomedical Engineering, University Medical Centre Groningen, University of Groningen, Antonius Deusinglaan 1, 9713 AV Groningen, The Netherlands; j.sjollema@umcg.nl

[4] LSRE-LCM—Laboratory of Separation and Reaction Engineering-Laboratory of Catalysis and Materials, Faculty of Engineering, University of Porto, Rua Dr. Roberto Frias, 4200-465 Porto, Portugal

[*] Correspondence: filipem@fe.up.pt; Tel.: +351-225-081-668

[†] These authors contributed equally to this work.

**Abstract:** Graphene and its functionalized derivatives have been increasingly applied in the biomedical field, particularly in the production of antimicrobial and anti-adhesive surfaces. This study aimed to evaluate the performance of graphene oxide (GO)/polydimethylsiloxane (PDMS) composites against *Staphylococcus aureus* and *Pseudomonas aeruginosa* biofilms. GO/PDMS composites containing different GO loadings (1, 3, and 5 wt.%) were synthesized and characterized regarding their morphology, roughness, and hydrophobicity, and tested for their ability to inhibit biofilm formation under conditions that mimic urinary tract environments. Biofilm formation was assessed by determining the number of total and culturable cells. Additionally, the antibacterial mechanisms of action of GO were investigated for the tested uropathogens. Results indicated that the surfaces containing GO had greater roughness and increased hydrophobicity than PDMS. Biofilm analysis showed that the 1 wt.% GO/PDMS composite was the most effective in reducing *S. aureus* biofilm formation. In opposition, *P. aeruginosa* biofilms were not inhibited by any of the synthesized composites. Furthermore, 1% (*w/v*) GO increased the membrane permeability, metabolic activity, and endogenous reactive oxygen species (ROS) synthesis in *S. aureus*. Altogether, these results suggest that GO/PDMS composites are promising materials for application in urinary catheters, although further investigation is required.

**Keywords:** graphene oxide; polydimethylsiloxane; *Staphylococcus aureus*; *Pseudomonas aeruginosa*; urinary catheters





## 1. Introduction

Since it was first isolated and characterized, graphene has been extensively studied for its unprecedented properties and possible applications [1]. Graphene is a single 2D layer of sp$^2$-hybridized carbon atoms covalently bonded, forming a hexagonal lattice structure [2] with great potential to be chemically functionalized [3]. This nanomaterial and its derivatives, such as graphene oxide (GO), are broadly recognized for their exceptional electronic [4], mechanical [5], thermal [6], and optical [7] properties. Furthermore, graphene materials have shown promising antibacterial properties against both Gram-negative and Gram-positive bacteria [8–10]. Even though the antibacterial mechanisms of action of these nanomaterials are not yet fully understood, they likely involve the combination of

physical and chemical interactions with bacterial cells, including cell membrane damage and endogenous reactive oxygen species (ROS) production [11–13].

Owing to their unique features, graphene-based materials have been used for multiple purposes in the biomedical context, for example, drug delivery [14], tissue engineering [15], bioimaging [16], biosensing [17], wound healing [18], and cancer therapy [19].

Among graphene derivatives, GO has been one of the most extensively studied [20]. This material is obtained through oxidation followed by the exfoliation of graphite, which provide 2D sheets of pristine graphene with sites of sp$^3$-hybridized carbon atoms and oxygenated groups [21]. Multiple studies have demonstrated GO's impressive antibacterial properties [22–24] and good biocompatibility [25,26], making it appealing for biomedical applications, in particular for the development of medical devices.

Indwelling urinary catheters (UCs) are standard medical devices that are inserted through the urethra or the lower abdominal wall to drain urine from the bladder. They are commonly used to manage conditions such as urinary retention or incontinence, as well as after surgery or other medical procedures. However, the use of UCs carries possible complications, including urinary tract infections (UTIs), bladder and kidney damage, and catheter blockage [27]. Up to 80% of UTIs in healthcare settings are associated with the use of UCs [28].

Most urinary tract devices are fabricated with silicone-based polymers, such as poly-dimethylsiloxane (PDMS) [29], which are very prone to bacterial colonization [30]. When a urinary catheter is inserted, it encounters a multitude of host urinary components, such as proteins and electrolytes, among other organic molecules present in the urinary tract [31]. The deposition of these components on the catheter surface forms a conditioning film that promotes bacterial attachment and subsequent biofilm formation. Uropathogens within biofilms are protected from host immune defenses as well as from the action of antimicrobial agents, thereby contributing to the persistence and spread of bacterial infections [32–34]. Ultimately, this leaves device removal as the only solution, which often carries nefarious consequences for patients [35]. As such, strategies targeting biofilm-specific processes and preventing biofilm formation on medical devices are required [36–38]. Among these new approaches, the employment of carbon-based materials, including graphene and carbon nanotubes, in medical devices has shown promising antibiofilm performance [39–42].

Thus, the main aim of this study was to produce and characterize PDMS-based surfaces with different GO loadings and assess their effects on the inhibition of early-stage biofilm development by *Staphylococcus aureus* and *Pseudomonas aeruginosa*—two common colonizers of UCs associated with UTIs [43,44]. The antibiofilm performance of the GO/PDMS surfaces produced against these bacteria was also compared with that of a previously optimized graphene nanoplatelets (GNP)/PDMS composite [39,45]. Furthermore, to gain a better understanding of the antibiofilm effect of GO/PDMS composites, the antimicrobial mechanisms of action of GO were scrutinized.

## 2. Materials and Methods

### 2.1. Synthesis of Graphene Oxide

GO was prepared using a modified Hummers' method, as described by Pedrosa et al. [46]. Initially, 240 mL of H$_2$SO$_4$ (Merck KGaA, Darmstadt, Germany) was gradually added under constant stirring (300 rpm) into a 2 L glass beaker containing 5 g of graphite (Sigma–Aldrich Chemie GmbH, Steinheim, Germany). To avoid overheating, the beaker was immersed in an ice bath. Subsequently, 5 g of NaNO$_3$ (AppliChem GmbH, Darmstadt, Germany) was carefully added to the mixture. After a 5–10 min interval, 30 g of KMnO$_4$ (Sigma–Aldrich Co., St. Louis, MO, USA) was added. Once adequate mixing was achieved, the mixture was removed from the ice bath and heated in an oil bath at 35 °C under similar stirring conditions (300 rpm) for 12 h. Before diluting with 1250 mL of distilled water and 35 mL of H$_2$O$_2$ (30% (*w*/*v*); VWR International, Leuven, Belgium), the mixture was cooled in an ice bath under stirring (400 rpm). Following this, the mixture was centrifuged (Eppendorf Centrifuge 5810R, Eppendorf, Hamburg, Germany) at 1744× *g* for 30 min to



remove excess acid. This resulted in the formation of a paste, which was filtered, washed with distilled water until a pH of 5 was reached, and dried overnight at 60 °C to obtain the finalized GO.

### 2.2. Surface Preparation

GO/PDMS composites were produced through a bulk mixing process by incorporating 1, 3, and 5 wt.% GO into Sylgard 184 Part A (Dow Corning, Midland, MI, USA), as explained by Oliveira et al. [39], and were designated as GO1/PDMS, GO3/PDMS, and GO5/PDMS, respectively. The GO/PDMS mixture was stirred at 500 rpm for 30 min and sonicated (Hielscher UP400S, Hielscher Ultrasonics GmbH, Teltow, Germany; 200 W, 12 kHz) for 1 h to ensure good dispersion of the nanomaterial into the PDMS matrix. Afterwards, to remove any remaining air bubbles, the composites were subjected to an ultrasonic bath (Ultrasons 3000514, JP Selecta, Barcelona, Spain) treatment for 30 min. Once concluded, Sylgard 184 Part B (Dow Corning, Midland, MI, USA) was added to PDMS/GO (at a 10:1 A:B ratio) and mixed. The GO composites were then deposited as a thin layer on top of glass (Vidraria Lousada, Lda, Lousada, Portugal; 1 × 1 cm slides) by spin coating (Spin150-v3.2, APT GmbH, Bienenbüttel, Germany; 1 min, 500 rpm ramp until 6000 rpm), allowing even spreading of the GO composites across the slides. Lastly, to allow the polymeric mixture to harden through heat, the coated coupons were maintained in the oven at 80 °C overnight. PDMS and 5 wt.% GNP/PDMS (designated as GNP/PDMS) surfaces were produced analogously and used as controls. Sonication, ultrasonic bath, spin coating, and heat curing steps were carried out under the same conditions to produce the control surfaces. Commercial GNP (Alfa Aesar, Thermo Fisher Scientific, Erlenbachweg, Germany) without any additional treatment were incorporated into the base elastomer, stirred, and subjected to all subsequent surface preparation steps to produce the GNP/PDMS composite [45].

### 2.3. Bacterial Strains and Culture Conditions

*Staphylococcus aureus* (SH1000) and *Pseudomonas aeruginosa* (PAO1) were chosen to assess the antibiofilm performance of the GO surfaces since these two bacteria are among the most commonly isolated from catheter-associated UTIs [47]. Bacteria were stored at –80 °C in Luria-Bertani Broth medium (LB; Thermo Fisher Scientific, Waltham, MA, USA) with 30% (*v/v*) glycerol. Prior to an experiment, each bacterium was spread on Plate Count Agar (PCA; Merck KGaA, Darmstadt, Germany) and incubated overnight (16–18 h) at 37 °C. Sterile LB medium was inoculated with colonies taken from those plates and incubated overnight at 37 °C in a shaker (Grant Bio™ PSU-10i, Fisher Scientific, Leicestershire, UK) at 160 rpm. After centrifugation for 10 min at 3100× *g* (Eppendorf Centrifuge 5810R, Eppendorf, Hamburg, Germany), cell pellets were resuspended in Artificial Urine Medium (AUM) [48], and bacterial suspensions with an optical density of 0.1 at 610 nm (around $1 \times 10^7$ cells·mL$^{-1}$) were prepared. According to Ramstedt et al. [49], surfaces for urinary tract devices should be tested using bacterial cell densities in the range of $10^7$–$10^9$ colony-forming units (CFU).mL$^{-1}$.

### 2.4. Surface Characterization

#### 2.4.1. Scanning Electron Microscopy (SEM)

To assess the morphology of coated surfaces, carbon adhesive tabs were used to mount the surfaces on aluminum stubs and they were sputter-coated with 5 nm chrome (ACE600, Leica Microsystems, Wetzlar, Germany). Samples were then imaged in a Zeiss Supra55 scanning electron microscope (Carl Zeiss Microscopy, Oberkochen, Germany) using the secondary electron detector at 3 kV.

#### 2.4.2. Optical Profilometry

To analyze the roughness of the synthetized surfaces, profilometry was performed using a non-contact profilometer (Proscan 2000, Scantron Industrial Products Ltd., Taunton, UK; scan area = 2 × 2 mm, step size = 0.01 μm, x-resolution and y-resolution = 0.01 μm).

The mean values and standard deviations of arithmetical mean height surface roughness ($S_a$) of the different surfaces result from quadruplicate measurements for each sample ($n = 4$). $S_a$ is representative of average surface roughness, as it corresponds to the absolute value of the difference in height of each point compared to the arithmetical mean of the surface [50]. To complement this analysis, an additional roughness parameter was determined (Table S1): the root mean square height ($S_q$), which is representative of the standard deviation of heights [51].

Quantitative data and representative 3D plots were obtained using a routine executed in MATLAB R2023a (The MathWorks, Inc., Natick, MA, USA).

### 2.4.3. Contact Angle Measurements

The hydrophobicity of the tested strain cells and the surfaces produced was determined through contact angle measurements, followed by the application of the van Oss approach [52].

To prepare bacterial substrata, *S. aureus* and *P. aeruginosa* cell suspensions containing about $1 \times 10^9$ cells·mL$^{-1}$ were filtered as indicated by Busscher et al. [53].

A contact angle meter (Dataphysics OCA 15 Plus, Filderstadt, Germany) was used to perform the measurements, using the sessile drop method. Water, $\alpha$-bromonaphthalene, and formamide (Sigma–Aldrich Co., St. Louis, MO, USA; liquid dispensed volume = 4 μL) were chosen as reference liquids in three independent assays. Measurements were performed instantaneously at room temperature. A minimum of three measurements were taken per bacterial substrata/surface for each independent assay ($n = 9$).

Free energy of interaction ($\Delta G$, mJ·m$^{-2}$) and free energy of adhesion ($\Delta G^{Adh}$, mJ·m$^{-2}$) were calculated as reported by Sousa-Cardoso et al. [45] to assess the hydrophobicity of the surfaces and the theoretical bacteria–surface thermodynamic affinity, respectively.

Considering a solid surface, negative free energy of interaction values ($\Delta G < 0$ mJ·m$^{-2}$) indicate that the surface is hydrophobic, while if $\Delta G > 0$ mJ·m$^{-2}$, the surface is hydrophilic [54].

Regarding the adhesion of the bacterial cells to a solid surface, it is thermodynamically favorable if $\Delta G^{Adh} < 0$ mJ·m$^{-2}$. Contrariwise, if $\Delta G^{Adh} > 0$ mJ·m$^{-2}$, adhesion is not favorable [55].

### 2.5. Biofilm Formation Assays

Biofilm formation assays were carried out on 12-well microtiter plates (VWR International, Carnaxide, Portugal) and duplicate coupons of each synthesized surface: 1 wt.% GO/PDMS (GO1/PDMS), 3 wt.% GO/PDMS (GO3/PDMS), 5 wt.% GO/PDMS (GO5/PDMS), and 5 wt.% GNP/PDMS (GNP/PDMS) and PDMS, as controls. First, 12-well polystyrene plates with transparent double-sided tape glued on each well and down-facing-coated glass square coupons were subjected to UV sterilization. Following that, the surfaces were fixed to the wells facing upward. After another round of UV exposure, 3 mL of *S. aureus* or *P. aeruginosa* bacterial suspension (prepared as described in Section 2.3) was added to each well. Control wells containing uninoculated AUM were used to ensure the sterility of the assays. The microtiter plates were then incubated under static conditions for 24 h at 37 °C. Three independent biofilm development assays were performed with two technical duplicates each ($n = 6$).

### 2.6. Biofilm Analysis

The number of biofilm total and culturable cells was quantified by flow cytometry and plate counts, respectively, whereas the biofilm amount was determined by crystal violet (CV) staining.

### 2.6.1. Bacterial Quantification

After biofilm formation for 24 h, the culture medium was carefully removed from the wells, and each coupon was placed into a 15 mL tube containing 2 mL of sterile saline

solution (8.5 g·L$^{-1}$ of NaCl). The tubes were vortexed for 3 min to promote the detachment of adhered bacteria from the surfaces. Adequate serial dilutions were performed in saline solution.

Total biofilm cells were quantified through flow cytometry (CytoFLEX V0-B3-R1, Beckman Coulter, Brea, CA, USA) as described by Sousa-Cardoso et al. [45]. For each surface, 10 μL of biofilm suspension was acquired at a flow rate of 10 μL·min$^{-1}$. CytExpert software (version 2.4.0.28, Beckman Coulter, Brea, CA, USA) was used to plot the bacterial populations, which were gated based on forward (FSC) and side-scatter (SSC) signals.

Appropriate dilutions of biofilm suspensions were also spread on PCA plates, which were incubated at 37 °C for 24 h. CFUs were enumerated to assess the number of culturable cells per coupon area (1 cm$^2$).

### 2.6.2. Biofilm Amount

The total mass of biofilms (biofilm matrix and cells) was quantified using the CV staining method. The medium was removed from the wells and the biofilms were stained with 3 mL of 0.1% (*v/v*) CV (Merck, Darmstadt, Germany) solution for 30 min. Then, they were rinsed twice with distilled water and allowed to dry naturally. Each biofilm sample was wiped thoroughly with an ethanol-soaked non-woven cloth and this material was dissolved in 3 mL of 1% (*v/v*) sodium dodecyl sulfate (SDS; VWR International, Lutterworth, UK) solution to extract the bound dye [56]. Absorbance was measured at 590 nm using a microtiter plate reader (SpectroStar Nano, Biogen Cientifica S. L., Madrid, Spain), and the values were used to estimate the biofilm index (*BI*) as follows (Equation (1)):

$$BI\ (\%) = \frac{R_0 - R_1}{R_0} \times 100 \tag{1}$$

where $R_0$ corresponds to the absorbance value of GO surfaces and $R_1$ is the absorbance value of PDMS (the control surface without GO).

### 2.7. Characterization of GO's Mechanisms of Action

The mechanisms of action of GO against *S. aureus* and *P. aeruginosa* were characterized using the flow cytometer. For each strain, a bacterial suspension containing around $10^7$ cells mL$^{-1}$ was treated with 1% (*w/v*) GO for 24 h at 37 °C. A non-treated cell suspension was maintained under the same conditions and used as a control. After this, the cell suspensions were centrifuged at 9715× *g* (Eppendorf 5418, Eppendorf AG, Hamburg, Germany) for 10 min. The supernatant was analyzed considering that free carbon materials would sediment and form a pellet [57].

Cell membrane potential, bacterial metabolic activity, and ROS production were assessed by staining the cells with bis-(1,3-dibutylbarbituric acid) trimethine oxonol (DiBAC$_4$(3); Sigma–Aldrich, Taufkirchen, Germany) at a final concentration of 0.5 μg·mL$^{-1}$ [58], 5(6)-carboxyfluorescein diacetate (5-CFDA; Sigma–Aldrich, Taufkirchen, Germany) at a final concentration of 5 μg·mL$^{-1}$ [59], and 2′,7′-dichlorofluorescein diacetate (DCFH-DA, Sigma–Aldrich, Taufkirchen, Germany) at a final concentration of 10 μM [60,61].

GO-treated and non-treated cell suspensions were stained for 30 min in the dark and analyzed by flow cytometry. Bacteria were gated based on FSC and SSC signals. Mean intensity of fluorescence (MIF) values at FL1 (fluorescence detector, 530 nm) were used to evaluate the effect of GO exposure on the aforementioned cell parameters.

### 2.8. Statistical Analysis

Descriptive statistics were used to determine the mean and standard deviation (SD) of the average roughness values, contact angles, number of biofilm total and culturable cells, and biofilm index. Differences between the number of biofilm cells were evaluated using independent-sample *t*-tests since the variables were normally distributed, according to the Shapiro–Wilk normality test. Statistically significant differences were considered for

*p*-values < 0.05. Data analysis was performed using the IBM SPSS Statistics version 29.0 for MacOS (IBM SPSS, Inc., Chicago, IL, USA).

## 3. Results and Discussion

### 3.1. Physicochemical Characterization of the Synthetized Surfaces

Since physicochemical surface properties, including morphology, roughness, and hydrophobicity, are known to impact bacterial adhesion and biofilm formation [62,63], the aforementioned parameters were analyzed for the five produced surfaces: PDMS, GNP/PDMS (surface controls without and with graphene, respectively), GO1/PDMS, GO3/PDMS, and GO5/PDMS (1, 3, and 5 wt.% GO incorporated into PDMS, respectively).

To assess the morphology of the surfaces at a nanometer level, SEM analysis was carried out (Figure 1). While PDMS displayed an overall homogeneous appearance (Figure 1a), all carbon-based composites showed a certain degree of heterogeneity across the analyzed surface area (Figure 1b–e). These results are in accordance with those of previous studies that assessed the morphology of graphene-based PDMS composites versus PDMS [39,45]. The GNP/PDMS surface (Figure 1b), while displaying visible irregularities, showed a considerably smoother morphology than GO/PDMS composites, which presented several small elevations on the surface of the coating, regardless of the concentration of GO incorporated into the polymeric matrix (Figure 1c–e), corresponding to agglomerates of graphene. It was also possible to observe that with an increase of GO loading from 1 to 5 wt.%, the number of visible clusters on the surfaces was greater. This is a consequence of the increment in the concentration of GO, which makes the uniform dispersion of this nanomaterial progressively more challenging in non-polar polymers, such as PDMS [64]. Furthermore, for the same loading of the carbon material (5 wt.%), SEM images suggest that there was a better dispersion of GNP (Figure 1b) than GO (Figure 1e) into the PDMS matrix. This is probably a result of the oxygen-containing functional groups that graphene acquires from the oxidation process of turning graphite into GO, which enable attractive interactions between GO nanosheets, namely strong van der Waals forces, promoting the observed clustering behavior [64].

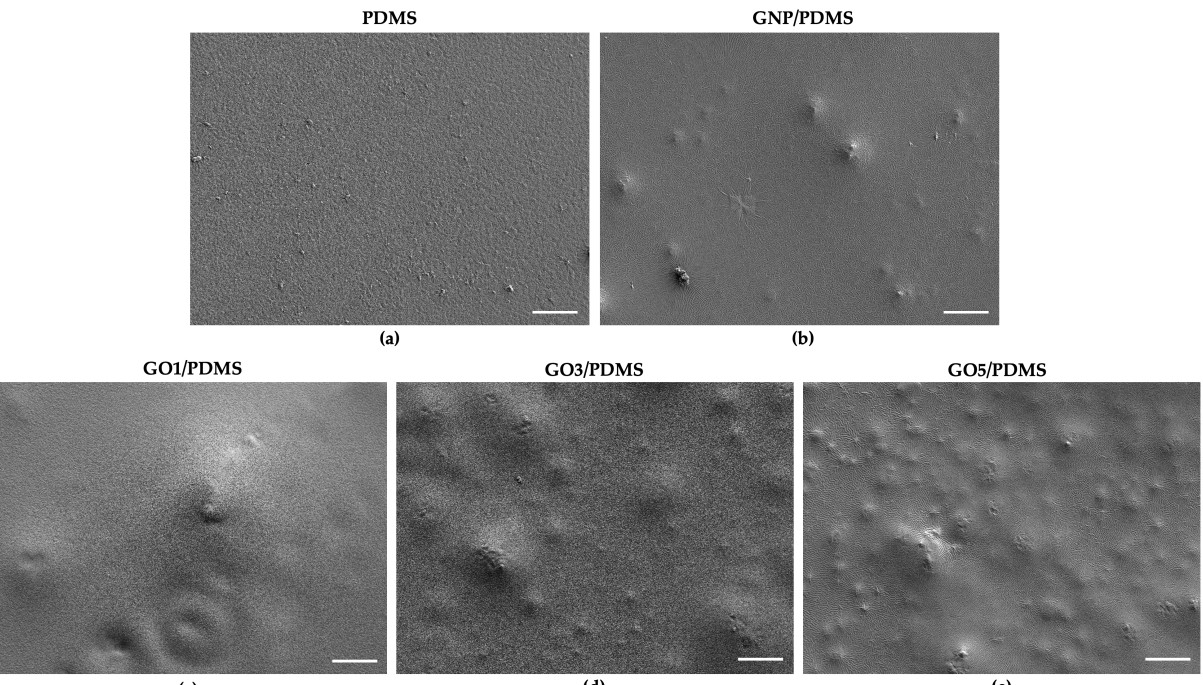

**Figure 1.** SEM images of PDMS (**a**), 5 wt.% GNP/PDMS (GNP/PDMS) (**b**), and GO/PDMS composites containing GO at different loadings: 1 (**c**), 3 (**d**), and 5 wt.% (**e**). The micrographs have a magnification of 500× and the white scale bars correspond to 50 μm.

The surface roughness was determined by profilometry analysis (Figure 2). Results showed a gradual increase in roughness from PDMS (Figure 2a) to GO5/PDMS (Figure 2e), i.e., PDMS stood out as the smoothest surface tested ($S_a$ = (47.45 ± 8.42) μm), while GO5/PDMS was the roughest composite, displaying an average roughness about 4 times greater than that of PDMS ($S_a$ = (207.96 ± 34.18) μm). Furthermore, at the same wt.% loading, the GO-based surface (Figure 2e) showed an average roughness around 143% greater than that of the GNP/PDMS surface (Figure 2b). Among the tested GO/PDMS surfaces, the most substantial increase in average roughness was found from GO3/PDMS (Figure 2d) to GO5/PDMS (Figure 2e)—an approximate 61% increase in $S_a$ was observed from GO3/PDMS to GO5/PDMS, compared to an 11% increase from GO1/PDMS to GO3/PDMS. This supports the previously mentioned difficulty of uniformly dispersing GO in non-polar polymers [64], especially for higher GO concentrations. Overall, these results corroborate the findings from the SEM analysis.

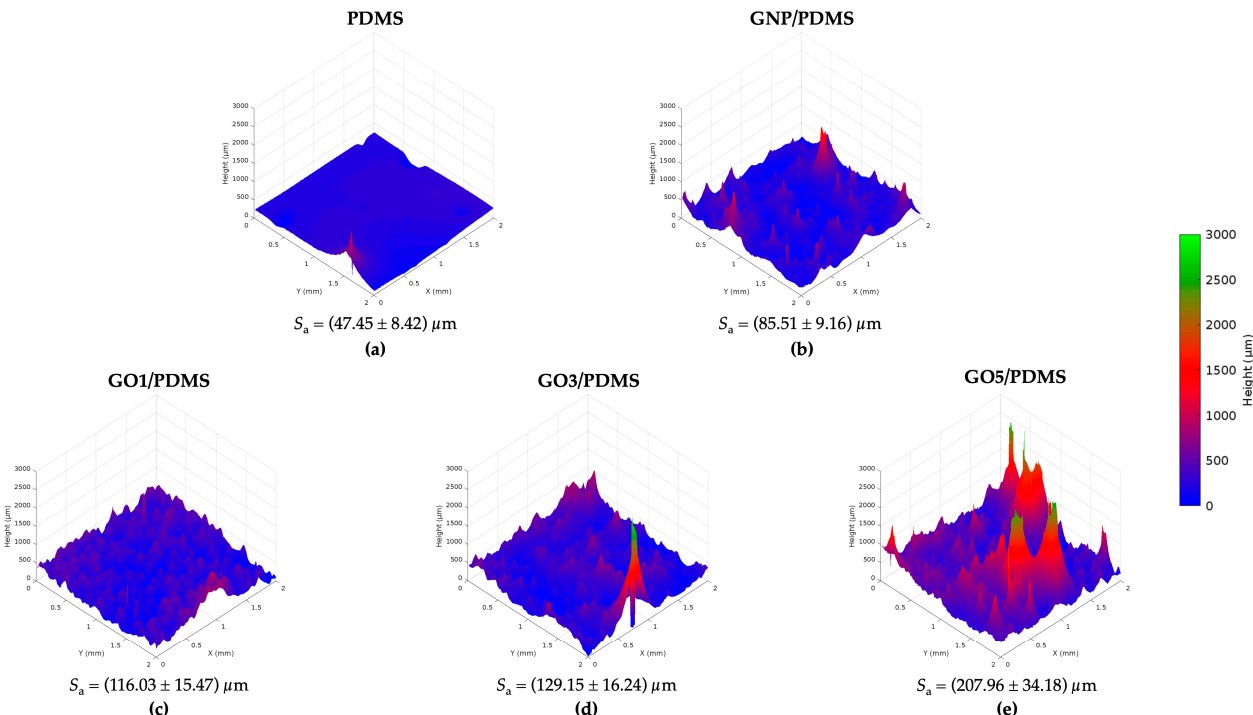

**Figure 2.** Representative profilometry plots of PDMS (**a**), 5 wt.% GNP/PDMS (GNP/PDMS) (**b**), and GO/PDMS composites containing GO at different loadings: 1 (**c**), 3 (**d**), and 5 wt.% (**e**). Average roughness ($S_a$) values for each surface are presented as mean ± SD and calculated based on the result of quadruplicate measurements for each sample.

The existence of graphene agglomerates on the surfaces can be beneficial for their antibiofilm performance since these clusters promote contact between the carbon nanomaterial and cells, thereby potentiating the composite antibacterial activity [65,66]. However, high surface roughness values are often associated with increased bacterial adhesion and biofilm formation [67–69]. As such, biofilm formation assays are essential to gain a better understanding of the interplay between these factors.

The hydrophobicity of the synthetized surfaces was assessed by contact angle ($\theta$) measurements and the determination of free energy of interaction ($\Delta G$) values (Table 1). All tested surfaces showcased contact angle values with water ($\theta_W$) higher than 90° and negative free energy of interaction values ($\Delta G$), which are indicative of a hydrophobic behavior. The three GO-based surfaces exhibited greater hydrophobicity than PDMS and GNP/PDMS, that is, lower $\Delta G$ values than those of PDMS and GNP/PDMS. This suggests that the interactions between GO nanosheets [63] are stronger than those with water [64]. Furthermore, the degree of surface hydrophobicity slightly increased with the increment

of GO loading. This increase was more noticeable from GO3/PDMS to GO5/PDMS; an approximately 1% increase in the absolute value of $\Delta G$ was observed from GO1/PDMS to GO3/PDMS compared to a 9% increase from GO3/PDMS to GO5/PDMS. Even though the differences in hydrophobicity of the three GO surfaces were not significant to make conclusions, these results suggest an association between the number of GO clusters on the surface/average surface roughness and the degree of hydrophobicity of the composites.

**Table 1.** Contact angles with the three reference liquids, and surface and bacteria hydrophobicity. Results are shown as mean $\pm$ SD.

| Sample | Contact Angle (°) | | | Hydrophobicity (mJ·m$^{-2}$) |
| --- | --- | --- | --- | --- |
| | $\theta_W$ | $\theta_F$ | $\theta_B$ | $\Delta G$ |
| **Surface** | | | | |
| PDMS | $114.3 \pm 1.4$ | $111.8 \pm 1.6$ | $88.3 \pm 3.8$ | $-63.1$ |
| GNP/PDMS | $115.8 \pm 2.1$ | $112.2 \pm 1.7$ | $89.8 \pm 3.7$ | $-67.5$ |
| GO1/PDMS | $122.3 \pm 4.1$ | $116.1 \pm 4.7$ | $81.5 \pm 3.3$ | $-75.8$ |
| GO3/PDMS | $116.2 \pm 1.0$ | $106.1 \pm 3.9$ | $77.0 \pm 4.4$ | $-76.8$ |
| GO5/PDMS | $118.1 \pm 2.2$ | $105.6 \pm 3.4$ | $77.4 \pm 4.9$ | $-83.7$ |
| **Bacteria** | | | | |
| *S. aureus* | $22.2 \pm 2.4$ | $42.4 \pm 4.0$ | $35.5 \pm 3.9$ | $44.6$ |
| *P. aeruginosa* | $20.3 \pm 2.3$ | $87.3 \pm 3.9$ | $34.1 \pm 3.8$ | $16.9$ |

Abbreviations: $\theta_W$—contact angle with water; $\theta_F$—contact angle with formamide; $\theta_B$—contact angle with α-bromonaphthalene; $\Delta G$—free energy of interaction.

Some studies suggest that highly hydrophobic surfaces are particularly effective in inhibiting bacterial adhesion [70,71]. However, bacterial surface hydrophobicity also plays a crucial role in the extent of biofilm formation [72,73].

According to water contact angle measurements, both tested bacteria presented a hydrophilic behavior (Table 1).

To further predict the thermodynamic affinity between the tested strains and the GO surfaces, the free energy of adhesion ($\Delta G^{Adh}$) of the microorganisms on each surface was determined (Table 2). In theory, the adhesion of *S. aureus* to the GO surfaces is thermodynamically favorable ($\Delta G^{Adh} < 0$), whereas the adhesion of *P. aeruginosa* is not favorable in any of the tested surfaces ($\Delta G^{Adh} > 0$). It is, however, important to consider that the formation of a conditioning film over the surfaces upon submersion in liquid media can have an impact on the electrostatic interactions between the bacteria and the surfaces they adhere to [74].

**Table 2.** Free energy of adhesion values of bacteria–surface interaction. Results are shown as mean $\pm$ SD.

| Surface | Bacteria–Surface Interaction ($\Delta G^{Adh}$, mJ·m$^{-2}$) | |
| --- | --- | --- |
| | *S. aureus* | *P. aeruginosa* |
| PDMS | 4.4 | 11.6 |
| GNP/PDMS | 3.0 | 10.7 |
| GO1/PDMS | $-2.5$ | 9.8 |
| GO3/PDMS | $-3.8$ | 10.1 |
| GO5/PDMS | $-6.6$ | 8.8 |

Moreover, biofilm formation itself can induce changes in the hydrophobicity of the substratum surfaces [75]. As such, these results, on their own, are not sufficient to predict the extent of biofilm development by the tested bacteria on each of the synthetized surfaces.

### 3.2. Antibiofilm Performance of the Synthetized Surfaces

Following surface characterization, early-stage biofilm development by *S. aureus* and *P. aeruginosa* on the five synthetized surfaces was assessed after incubation at 37 °C for 24 h under static conditions. This mimics the colonization of the extraluminal side of UCs which is exposed to quasi-static urine in the bladder [49]. Biofilm formation is a dynamic process that involves several steps occurring in tandem to form a highly structured community of microbial cells embedded in a self-produced matrix. Generally, early-stage biofilm formation occurs within the first 24 h and comprises adhesion and aggregation of microbial cells on the surface [76,77]. Monitoring the early-stage of biofilm growth is of paramount importance as it allows the design of strategies for the prevention of mature biofilms.

Biofilm development was analyzed by total and culturable cell enumeration obtained through flow cytometric analysis and CFU counting, respectively (Figure 3). For *S. aureus*, results revealed that the PDMS and GNP/PDMS surfaces displayed a significantly higher number of biofilm total cells than GO-based surfaces ($p < 0.05$, Figure 3a). Among the GO/PDMS composites, GO1/PDMS was the most promising surface, showing a reduction of approximately 49% in the number of total biofilm cells in comparison to the PDMS control surface (on average, $6.69 \times 10^5$ cells·cm$^{-2}$ on GO1/PDMS and $1.58 \times 10^6$ cells·cm$^{-2}$ on PDMS, $p < 0.05$). These results demonstrate the antibiofilm activity of GO1/PDMS surface against *S. aureus*. Regarding biofilm culturable cells, PDMS and GNP/PDMS displayed, on average, a greater number of cells compared to the GO/PDMS surfaces (Figure 3b). In fact, both GO1/PDMS and GO3/PDMS significantly reduced the number of culturable cells by 55% and 32% ($p < 0.05$), respectively, in comparison with the PDMS surface (on average, $5.91 \times 10^5$ cells·cm$^{-2}$ on GO1/PDMS, $8.91 \times 10^5$ cells·cm$^{-2}$ on GO3/PDMS, and $1.31 \times 10^6$ cells·cm$^{-2}$ on PDMS). These results support the antimicrobial activity of GO1/PDMS surface against *S. aureus* biofilms. Regarding *P. aeruginosa*, there were no significant differences in the number of biofilm cells grown on each of the five tested surfaces (Figure 3c,d).

These results are according to those of previous studies that reported a greater antibacterial effect of GO against Gram-positive bacteria than Gram-negative bacteria [10,24,78,79]. Gram-positive bacteria, such as *S. aureus*, possess a cytoplasmic membrane surrounded by a peptidoglycan layer, whereas Gram-negative bacteria, such as *P. aeruginosa*, possess a complex cell envelope composed of a plasma membrane, a peptidoglycan cell wall, and an outer membrane mainly made up of lipopolysaccharide (LPS) [80]. This outer membrane is considered to largely contribute to the stiffness and strength of Gram-negative cells [81,82] and may hinder one of the main postulated mechanisms of action of graphene materials, cell membrane disruption. In fact, Atomic Force Microscopy (AFM) results reported by Di Giulio et al. [78] showed that although GO was capable of trapping *S. aureus*, this effect was not detected for *P. aeruginosa*. This suggests that GO can entrap Gram-positive bacteria by interacting with the peptidoglycan layer, which, in Gram-negative bacteria, is protected by the LPS outer membrane, with whom GO is reported to have predominantly repulsive interactions [83].

Furthermore, these results suggested that adhesion and biofilm formation processes were influenced not only by the properties of the surfaces (e.g., roughness and hydrophobicity) but also by the tested bacterium itself.

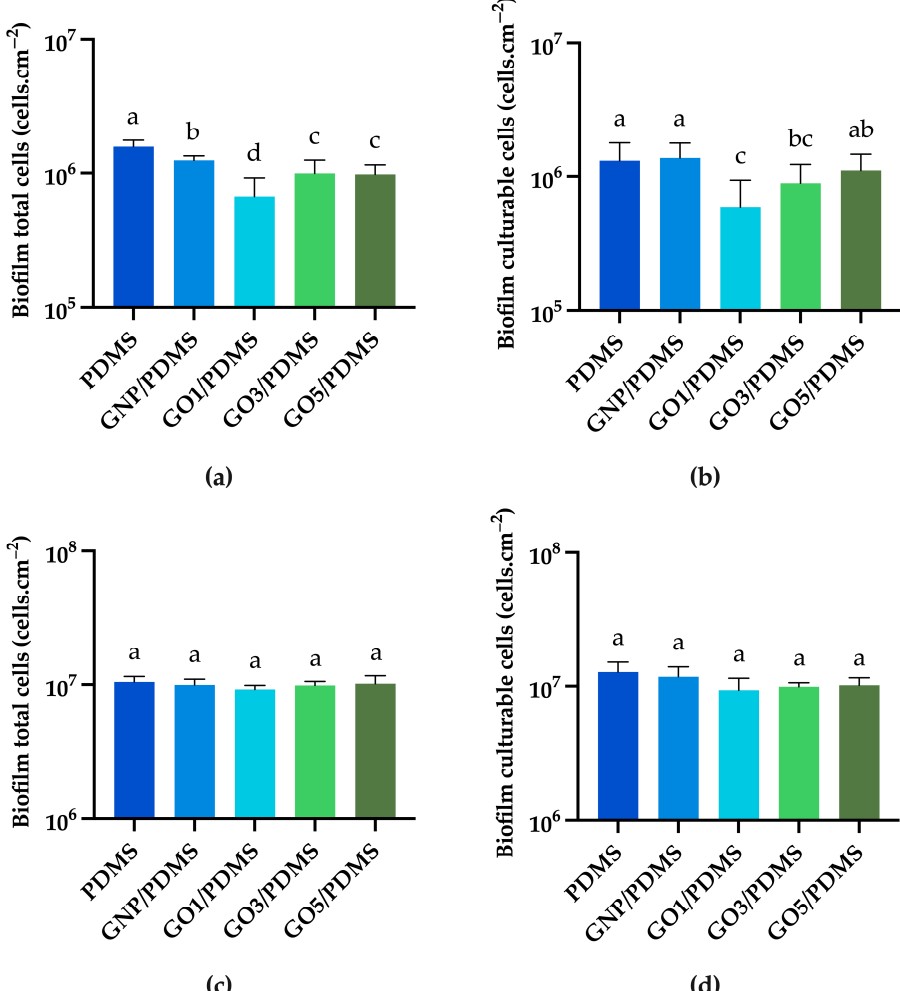

**Figure 3.** *S. aureus* (**a**,**b**) and *P. aeruginosa* (**c**,**d**) biofilm development on different surfaces: PDMS (■), GNP/PDMS (■), GO1/PDMS (■), GO3/PDMS (■), and GO5/PDMS (■). The parameters analyzed refer to biofilm total cells (**a**,**c**) and culturable cells (**b**,**d**) per $cm^2$. Mean values ± SD from three biological assays with two technical replicates each are represented. For each graph, different lowercase letters (a, b, c, and d) indicate significant differences between the number of cells per area ($p < 0.05$).

To complement the information provided by the sessile cell number (total and culturable cells), the amount of biofilm (matrix plus cells) formed by *S. aureus* and *P. aeruginosa* on the PDMS and the two most promising antibiofilm surfaces (GO1/PDMS and GO3/PDMS) was estimated using the standard crystal violet method [56]. The absorbance values given by this method enabled the determination of the biofilm formation index (Figure 4) which reflects the ability of GO composites to inhibit biofilm development compared to PDMS, the control surface without graphene. Both surfaces decreased the amount of *S. aureus* biofilm formed by a similar percentage, but enhanced *P. aeruginosa* biofilm development compared to PDMS. Given that the number of total cells was shown to be identical between surfaces for the *Pseudomonas* strain (Figure 3c), the increase in the biofilm amount compared to the control surface revealed by the CV method can be indicative of a greater amount of extracellular polymeric substances (EPS) formed.

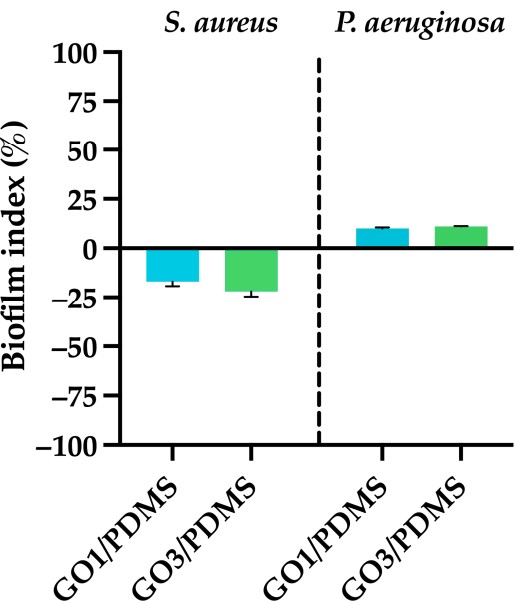

**Figure 4.** Biofilm formation index of *S. aureus* and *P. aeruginosa* on GO1/PDMS (■) and GO3/PDMS (■). These values were determined using the crystal violet staining method and considering the PDMS as the reference surface. Mean values ± SD from six replicates are represented.

Apart from the results obtained for the GO-based composites, contrary to what was expected, GNP/PDMS did not exhibit enhanced antibiofilm properties compared to the PDMS control for both *S. aureus* and *P. aeruginosa* under the tested conditions. These findings highlight the relevance of functionalization as a promising approach to increase the antimicrobial activity of graphene, as well as the importance of testing antibiofilm surfaces under conditions mimicking those of realistic settings.

### 3.3. Mechanisms of Action of GO

To unveil the mechanisms of action of GO against the tested bacteria, *S. aureus* and *P. aeruginosa* cells were exposed to 1% (*w/v*) GO for 24 h. After this period, they were stained with DiBAC$_4$(3), 5-CFDA, and DCFH-DA, and analyzed by flow cytometry (Figures 5 and 6 for *S. aureus* and *P. aeruginosa*, respectively). Non-treated cells were used as control.

*S. aureus* cells exposed to GO and stained with DiBAC$_4$(3) displayed a higher mean intensity of fluorescence (MIF) than non-treated cells (4-fold higher, Figure 5a,b). As DiBAC$_4$(3) is only able to enter depolarized cells, these data indicate that GO exposure triggered cell membrane depolarization. By staining *S. aureus* with 5-CFDA, it was possible to observe that GO substantially increased the metabolic activity of treated cells (3-fold higher MIF than non-treated cells, Figure 5c,d), which suggests that bacteria are reprogramming their metabolism to adapt to an unfavorable environment [65]. In turn, *S. aureus* exposed to GO and stained with DCFH-DA presented a higher MIF than non-treated cells (2-fold higher, Figure 5e,f). This result demonstrated that GO exposure led to ROS production. It is known that changes in metabolism as a response to stress may be associated with endogenous ROS production [84]. Overall, these findings suggest that GO targets the bacterial cell membrane, inducing depolarization, and leading to ROS production, which is in accordance with the hypothesized mechanisms of action of graphene materials [85] and corroborates the antimicrobial activity of GO/PDMS surfaces against *S. aureus*.

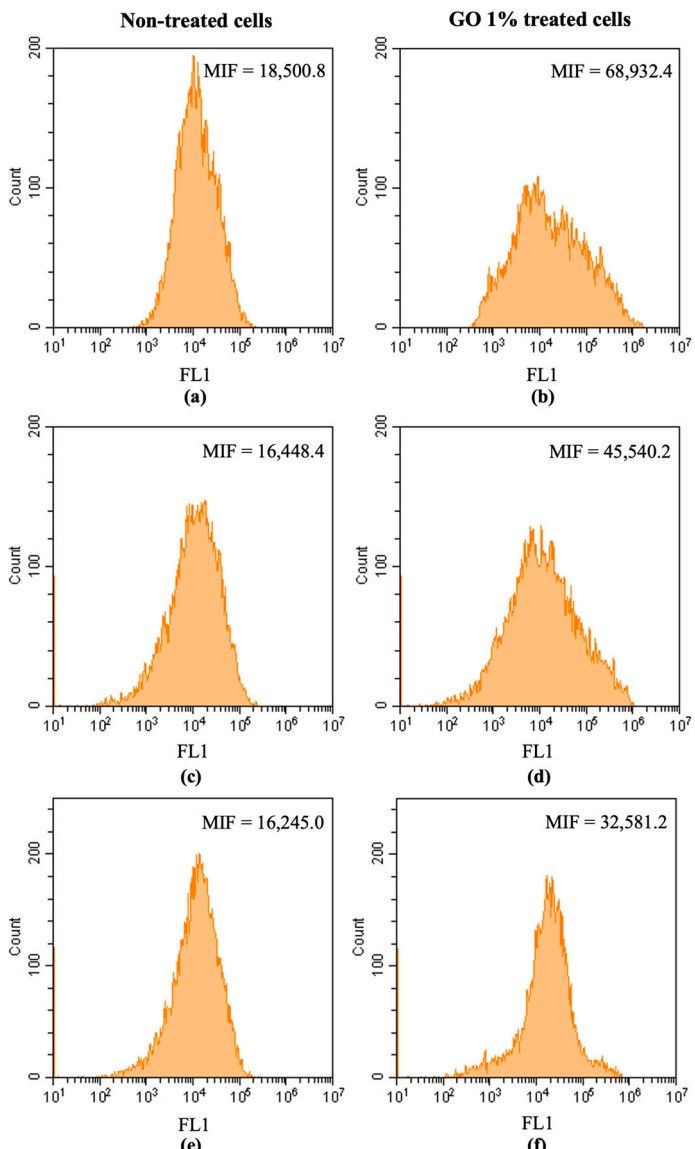

**Figure 5.** Flow cytometry histograms obtained for *S. aureus* non-treated and treated with 1% (*w/v*) GO stained with DiBAC$_4$(3) (**a,b**), 5-CFDA (**c,d**), and DCFH-DA (**e,f**), respectively. Results are shown as the mean intensity of fluorescence (MIF).

*P. aeruginosa* cells exposed to GO and stained with DiBAC$_4$(3) did not reveal changes in membrane permeability (similar MIF values for non-treated and GO-treated cells, Figure 6a,b). However, *P. aeruginosa* cells exposed to GO and stained with 5-CFDA and DCFH-DA displayed higher MIF values than non-treated ones (2-fold higher MIF, Figure 6c–f), indicating that GO exposure increased cell metabolic activity and ROS production. These findings suggest that, although GO created a hostile environment for the cells, it was not able to inactivate *P. aeruginosa* under the tested conditions. These results corroborate the lack of efficacy of GO/PDMS composites against *P. aeruginosa* biofilms.

All in all, by significantly hindering *S. aureus* initial adhesion over 24 h, GO1/PDMS shows great potential for employment in UCs and reduce the failure rate of these medical devices.

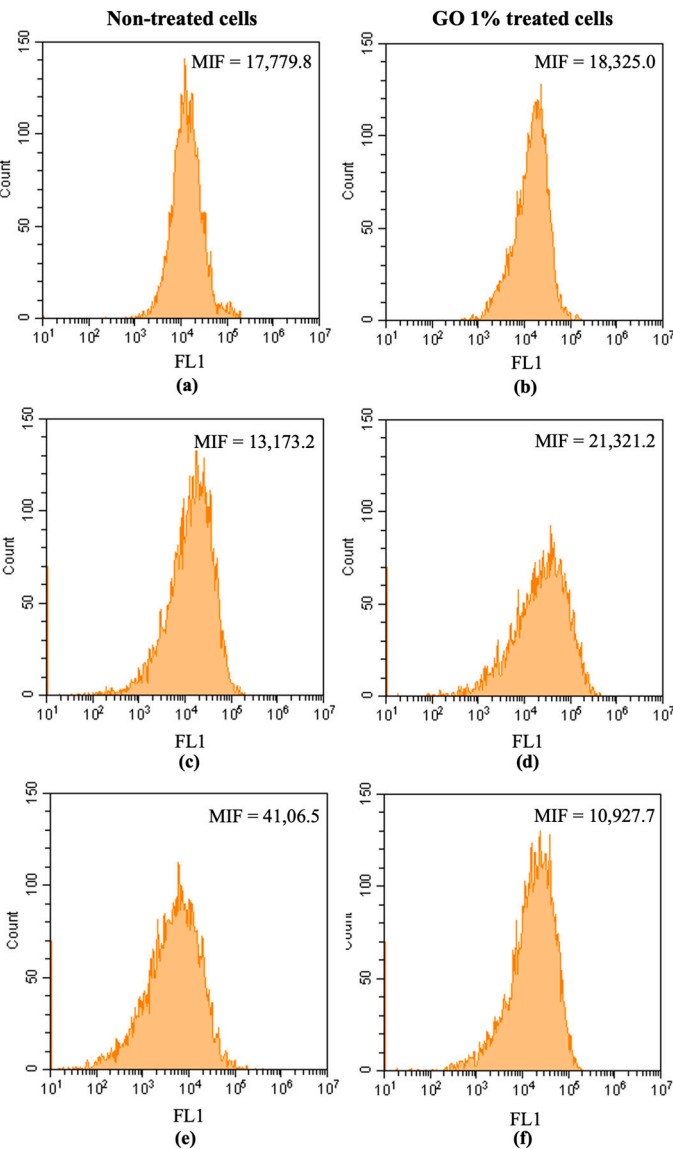

**Figure 6.** Flow cytometry histograms obtained for *P. aeruginosa* non-treated and treated with 1% (*w/v*) GO, stained with DiBAC$_4$(3) (**a,b**), 5-CFDA (**c,d**), and DCFH-DA (**e,f**), respectively. Results are shown as the mean intensity of fluorescence (MIF).

## 4. Conclusions

In this study, GO-based surfaces were investigated in terms of their physicochemical properties as well as their effectiveness in inhibiting *S. aureus* and *P. aeruginosa* biofilm formation under conditions that mimic urinary tract environments. The incorporation of GO into a PDMS matrix resulted in composites with increased roughness and hydrophobicity compared to PDMS and GNP/PDMS. Biofilm analysis showed that the composite with a 1 wt.% loading was effective in reducing *S. aureus* biofilm formation. In opposition, GO/PDMS composites were not able to significantly reduce *P. aeruginosa* biofilms. A comprehensive analysis of GO's mechanisms of action revealed that this carbon material targets the *S. aureus* cell membrane and induces ROS production, while in *P. aeruginosa*, it only induces ROS production without inactivating the cells. Although these results demonstrate the potential of GO-based composites to coat urinary catheters, further research is required to improve their activity against Gram-negative uropathogens.

**Supplementary Materials:** The following supporting information can be downloaded at https://www.mdpi.com/article/10.3390/coatings13081324/s1, Table S1: Average roughness ($S_a$) and root mean square height ($S_q$) values determined for each of the tested surfaces. Results are presented as mean $\pm$ SD.

**Author Contributions:** Conceptualization, R.T.-S., L.C.G. and F.J.M.; methodology, S.B., F.S.-C. and R.V.; formal analysis, S.B., F.S.-C., R.T.-S and L.C.G.; resources, J.S., O.S.G.P.S. and F.J.M.; data curation, S.B., F.S.-C., R.T.-S and L.C.G.; writing—original draft preparation, S.B. and F.S.-C.; writing—review and editing, R.T.-S., L.C.G., J.S., O.S.G.P.S. and F.J.M.; supervision, F.J.M. All authors have read and agreed to the published version of the manuscript.

**Funding:** This work was financially supported by LA/P/0045/2020 (ALiCE), UIDB/00511/2020, UIDP/00511/2020 (LEPABE), UIDB/50020/2020, and UIDP/50020/2020 (LSRE-LCM) funded by national funds through FCT/MCTES (PIDDAC); project NanoCAT (PTDC/CTMCOM/4844/2020), supported by national funds through the FCT/MCTES (PIDDAC); project 2SMART (NORTE-01-0145-FEDER-000054), supported by Norte Portugal Regional Operational Programme (NORTE 2020), under the PORTUGAL 2020 Partnership Agreement through the European Regional Development Fund (ERDF), and project SurfSAFE supported by the European Union's Horizon 2020 Research and Innovation Programme under grant agreement number 952471. R.T.-S. acknowledges the receipt of a junior researcher fellowship from project PTDC/CTM-COM/4844/2020 (NanoCAT). O.S.G.P.S. thanks FCT for the financial support of her work contract through the Scientific Employment Stimulus—Institutional Call—CEECINST/00049/2018.

**Institutional Review Board Statement:** Not applicable.

**Informed Consent Statement:** Not applicable.

**Data Availability Statement:** The data presented in this study are available from the corresponding author upon request.

**Conflicts of Interest:** The authors declare no conflict of interest.

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
