# Peer review of "Production and Characterization of Graphene Oxide Surfaces against Uropathogens"

_coatings, doi:10.3390/coatings13081324_

Round 1

Reviewer 1 Report

The article entitled Antibiofilm Effect of Graphene Oxide Surfaces Against Uropathogens concerns an important problem, i.e., bacterial infection in the urinary tract caused by the presence of urological catheters. Shows the potential to increase the resistance to biofilm formation of one of the most used materials, i.e., PDMS, by the addition of GO. Even that the antibacterial properties of the composite films is rather minor, it can indicate the direction of further material development. The manuscript is well written, with proper description of the methods and results including also valid scientific discussion.

Nevertheless, in the contact angle description it should be included the volume of the drop, and the time at which contact angle was measured, instant, after 1s, 5s?

In the text it is mentioned that the agglomerates can be beneficial for antibacterial properties, however the is a consideration about repeatability of the surface roughness then. Usually, the formation of filler agglomerates is a very statistical process, which can lead to films with different roughness despite the same amount of filler. On how many samples were roughness measurements performed and what were the differences between the samples?

(contact angle) The discussion and drawn conclusions are an overinterpretation of the results. Taking into account the standard deviation practically there are no differenced between water contact angle values. So the conclusions about interactions and so on are doubtful. Also, in case of rough, heterogeneous surface rather the dynamic than static contact angle should be done. It is highly possible that the differences in values of contact angle are due to different roughness of samples. Still the results are valuable, but this part (lines 300-310) should be improved.

Reviewer 2 Report

The manuscript of Belo et al. analyzes the effect of GO/PDMS composites, prepared by the authors, on the biofilm formation of one Gram-positive and one Gram-negative bacteria. Generally, the manuscript is well written, and the surface materials are well characterized, but the obtained results are not very promising. The main result is the reduction of the number of total biofilm cells and biofilm culturable cells on the surface of GO1/PDMS. Taking into consideration the high error of the reported measures,  the result should be further validated using at least a different technique for biofilm evaluation, e.g., classical Crystal violet staining or confocal laser scanning microscopy (if applicable). Moreover, no hypothesis about the different behavior of the other analyzed materials concerning biofilm formation has been made, and the correlation between biofilm formation and the analyzed material properties has not been discussed.

Minor points:

The preparation of GNP/PDMS should be briefly explained to highlight the differences with the new materials.

Lane 348, LPS are the main component of the outer membrane, not the only one

Reviewer 3 Report

The title (Antibiofilm Effect of Graphene Oxide Surfaces Against Uropathogens). It should be preparation, development characterization of (such and such).

The claim of having antibiofilm is totally irrelevant. There is no anti-biofilm activity of those prepared surfaces. and the SEM agglomerate may suggest higher adhesion due to their roughness, also, encrtustation which is formation of calcium and magnesium salts if pH increased will worsen the hypothesis.  

The 57% similarity report makes me rise some questions, the method in contact angle was adapted without rephrasing,  

Als very nice data without connecting the results with their application, that is the difference in initial steps of bacterial adhesion.  

I found it important to the interface related field of chemistry and physics, the biologic part needs addressing: The choice of the microorganisms, the high load of adhered bacteria. The lack of the physicochemical properties on the effect of adhesion.  

They need to think of a control that show higher adhesion of bacteria, or different contact angle because they are similar. A typical rubber Foley catheter, or silicone mold may do it.  

Line 95: rephrase . before adding the next reagetn, specify

Wording like bare (line247) better be more scientific

Line 245: explain why those designated parameters are crucial

Line 291: Those agglomerate could worsen encrustation which is harmful as musch as biofil formation

Scientific language is advised

Round 2

Reviewer 2 Report

In my opinion, the paper can be published in the present form

Author Response

The authors are grateful for the Reviewer's comment.